# A Centrifugal Pump Fault Diagnosis Framework Based on Supervised Contrastive Learning

**DOI:** 10.3390/s22176448

**Published:** 2022-08-26

**Authors:** Sajjad Ahmad, Zahoor Ahmad, Jong-Myon Kim

**Affiliations:** 1Department of Electrical, Electronic and Computer Engineering, University of Ulsan, Ulsan 44610, Korea; 2PD Technology Cooperation, Ulsan 44610, Korea

**Keywords:** centrifugal pumps, convolutional encoder, contrastive learning, fault diagnosis, deep learning

## Abstract

A novel intelligent centrifugal pump (CP) fault diagnosis method is proposed in this paper. The method is based on the contrast in vibration data obtained from a centrifugal pump (CP) under several operating conditions. The vibration signals data obtained from a CP are non-stationary because of the impulses caused by different faults; thus, traditional time domain and frequency domain analyses such as fast Fourier transform and Walsh transform are not the best option to pre-process the non-stationary signals. First, to visualize the fault-related impulses in vibration data, we computed the kurtogram images of time series vibration sequences. To extract the discriminant features related to faults from the kurtogram images, we used a deep learning tool convolutional encoder (CE) with a supervised contrastive loss. The supervised contrastive loss pulls together samples belonging to the same class, while pushing apart samples belonging to a different class. The convolutional encoder was pretrained on the kurtograms with the supervised contrastive loss to infer the contrasting features belonging to different CP data classes. After pretraining with the supervised contrastive loss, the learned representations of the convolutional encoder were kept as obtained, and a linear classifier was trained above the frozen convolutional encoder, which completed the fault identification. The proposed model was validated with data collected from a real industrial testbed, yielding a high classification accuracy of 99.1% and an error of less than 1%. Furthermore, to prove the proposed model robust, it was validated on CP data with 3.0 and 3.5 bar inlet pressure.

## 1. Introduction

Centrifugal pumps (CPs) are vital to various technical processes such as power generation, chemical processes, air conditioning, manufacturing, heating, and cooling of engines [1]. According to a survey of European organizations, machines driven by electric motors consumes approximately 65% of the grid energy, and CPs consume 80% of the total energy [2]. CPs are diverse in use, inexpensive, easy to construct, and reliable to operate. However, they can fail unexpectedly, which can lead to problems, i.e., economic loss, energy loss, expensive repairs, danger to the life of the operating staff, and lengthy downtimes [3]. Thus, to prevent their failure and avoid these severe consequences ensuring reliability, a health-monitoring system for CPs is necessary.

There are several ways to monitor the health of CPs. Among them, the condition-based monitoring system (CBM) is the most effective. The CBM is based on data collected from the machine under different conditions; thus, it is helpful to extend the runtime of a machine with low cost [4]. Hence, in this study we also used a condition-based monitoring system for fault diagnosis in CPs. According to research conducted on CPs, there are two types of faults, i.e., hydraulic and mechanical. These types of faults are dependent on each other, but mechanical faults are more frequent [5]. Thus, to keep a CP healthy, mechanical faults, such as impeller faults, mechanical seal holes, and scratches, need to be identified early. Therefore, in this study we considered only the mechanical fault diagnosis in CPs. Mechanical faults have a huge impact on the vibration signals obtained from a CP; these faults make the vibration signals non-stationary and impulsive, which draws attention to the analysis of these vibrations for fault identification [6]. Due to faded fault impulses and a significant amount of noise in the signals, fault-related information is often unclear and masked, so it is necessary to pre-process the vibration sequences and extract useful fault-related information from the signal. The main techniques used for signal processing are analyses in time (TD), frequency (FD), and time–frequency domains (TFD).

TD-extracted features, such as RMS, peak value, variance, etc., can be useful in fault identification of bearings and gearbox; however, in CPs, there are unexpected changes in the height (amplitude) of the signals, causing these TD-extracted features to provide very little discriminating information. However, the discriminant features can still be extracted using the FD. The FD can address the issue of impulses in the signal by displaying a peak in the specific frequency spectrum. The FD analysis of stationary and simple vibration signals is useful for fault identification; however, in this study, we considered a vibration signal with a non-stationary and complex nature. Thus, the FD analysis tool, Fourier transform, could not be used to extract features from these non-stationary and complex signals. However, we could use FD analysis in shape of spectral kurtosis to identify the transients in the CP vibration signal, though fault features interpretation would remain complex [7]. To overcome the shortcomings of the FD, such as its limitation of applicability to only stationary signals and its complexity, researchers have used several TFD techniques for the detection of faults using non-stationary signals. Peng et al. [8] used a general parametrized time–frequency transform for feature extraction from a non-stationary vibration signal. Kang et al. [9] used a time-changing and multi-problem solving solution envelope analysis for fault diagnosis in bearings. Alexander Prosvirin et al. [10] used empirical mode decomposition (EMD) for rub-impact fault identification in blades utilizing non-stationary vibration signals. The TFD analyzes the non-stationary and complex signals with better precision; however, it still depends on discriminant feature extraction for the proper characterization of mechanical faults. A better technique to solve these issues is spectral analysis. The spectral analysis tool, kurtogram, which was used in various studies [3] has proven to be effective and the best among all the techniques used to deal with non-stationary and impulsive signals for fault detection. Based on its effectiveness in fault diagnosis, in this study we also used kurtogram to differentiate between different frequencies based on colors assigned to them in the kurtogram image of the vibrations signal.

For fault diagnosis, when the features are extracted, they are usually handed over to machine learning (ML) algorithms, i.e., SVM [11], K-NN [12,13], or naïve Bayes classifiers, for classification. The mentioned ML methods result in a satisfactory classification, especially, SVM proved to provide a high accuracy in research conducted in [14]. In addition, a significant amount of research has been performed on artificial intelligence-based fault diagnosis in pumps [15]. However, the related algorithms classify data based on manually extracted features, which may not be the best for the classification of faults as it requires expertise in manual data analysis. This strategy is also very time-consuming even for experienced analyzers. It is necessary to extract features that provide the best discrimination for data from different classes. Therefore, to avoid the complications associated with conventional ML algorithms and extract the useful features, deep learning (DL) models have been used [16,17]. Deep learning is useful in fault diagnosis because of its fault features extraction and classification performance. Deep learning, with its inherent property of autonomous feature extraction, solves both the issues linked to ML models, i.e., autonomous important features extraction and time-consuming manual data analysis. Some of the deep learning techniques used for fault diagnosis include a convolutional neural network (CNN) [3] and autoencoders [18]. The CNN is the most popular DL technique used for feature extraction [19], and autoencoders are convenient for feature extraction because of their ability to compress data and form latent coding. Guo et al. [20] utilized a deep belief network (DBF) for fault diagnosis in bearings. Based on previous studies, all the deep learning techniques showed a prominent and satisfactory performance in feature extraction and classification. However, it is still possible to increase the accuracy and robustness of these models. Thus, to improve the DL model for fault classification, we introduced a supervised approach using contrastive learning, which allowed learning the contrast in data fed to the DL model. Supervised contrastive learning uses the supervised contrastive loss that pulls together samples of the same class while pushing apart samples belonging to different classes, providing better discriminance in data than other models. Therefore, supervised contrastive learning outperforms other loss functions in terms of improving the accuracy and robustness of a DL model [21]. In this work, we present a supervised contrastive learning-based DL framework for mechanical fault diagnosis in CPs. We used a CE to produce the vector representations of the input images and applied supervised contrastive loss to learn the representations of images and bring together those belonging to the same class, while moving apart those coming from different classes. A linear classifier was built with a CE having a fully connected layer accompanied by a SoftMax layer. First, the CE was pretrained without a classifier to optimize the supervised contrastive loss. The CE with supervised contrastive loss learnt the contrasting representations corresponding to the respective class; then, for the classification of CP faults, the weights of the CE were kept frozen, and a classifier (linear) was trained using the same weights of the already trained CE, while the weights of the hidden and SoftMax layers of the classifier were optimized for fault identification. The major contributions of this study are:In order to find the impactful and discriminant parts in vibration signals, we computed fast kurtograms of the time series vibration signals. The kurtograms displayed the fault-related transients in collected signals well.To address the issues of conventional feature extraction methods, a convolutional encoder was introduced to produce a latent space with the help of its compressing power.To make the deep learning models more accurate and robust, a supervised contrastive loss function was employed, which clearly outperformed the conventional loss functions. Based on the data contrast, the classifier carried out the classification task, completing the process of faults diagnosis.

The rest of the work is arranged as follows:

Section 2 presents the proposed method, Section 3 contains the experimental testbed setup and data acquisition, Section 4 presents the technical background, Section 5 describes the results and performance, and Section 6 concludes the manuscript.

## 2. Proposed Method

Figure 1 shows the CP fault diagnosis framework, comprising four main steps.

IThe vibration signals under different CP conditions were collected using a data acquisition system.IIThe kurtograms were computed from the collected vibration signals and displayed the frequency changes in different sub-bands of the signals in the form of an image pattern used for fault diagnosis.IIINext, the kurtograms images were fed to the CE with a supervised contrastive loss function. This CE was pretrained for supervised contrastive loss optimization that segregated the data of a corresponding label using data contrast.IVAfter the CE learnt the contrastive features of a corresponding label, the CE was kept frozen; meanwhile, a linear classifier was trained, and the classifier accomplished the task of classification.

## 3. Experimental Setup and Data Acquisition

Figure 2 depicts the self-developed experimental test rig comprising a PMT-4008 CP driven by a motor with a power of 5.5 kw, a control panel that changed the flowrate, speed, temperature, and water supply, a display, and a switch. Two water tanks were used to keep a net positive suction head at the inlet of the pump. The water tanks were placed high enough to normally operate the CP. Steel pipes with pressure gauges and valves were used to connect the CP to the tanks. The schematized the testbed is displayed in Figure 3. The CP was constantly run at 1733 rpm during the collection of the vibration signals. The vibration data were obtained using several accelerometers; two were fixed on the pump case, one was installed near the mechanical seal (MS), and one was fitted near the impellers, using adhesives. Every accelerometer had its own channel to record the CP vibration data. A National Instruments 9234 device was used to digitize the vibration data. Table 1 presents the details of the accelerometer and the digitization of the collected CP vibration signals.

For the collection of the data, the CP was run for 300 s at 25.6 kHz sampling frequency. A sum of 1200 samples with 25,600 sample length were collected under one normal and three faulty conditions, at an inlet pressure of 3.0 bar. The same process was repeated applying a 3.5 bar pressure at the inlet. Further details of the experimental setup are reported in [3].

This study analyzed the CP data collected under four varying operating conditions: a normal condition, an MS scratch fault, an MS hole fault, and an impeller defect. The description of different fault conditions is provided below.

### 3.1. Mechanical Seal Scratch Fault

In this study, an MS of 38 mm diameter was used. A scratch having a diameter of 2.5 mm and a depth of 2.8 mm was created in the seal for a seal scratch fault. Figure 4 presents the actual fault condition.

### 3.2. Mechanical Seal Hole Fault

For the hole fault, the mechanical seal’s diameter was 38 mm. A hole with a diameter and a depth of 2.8 mm was made in the revolving part of the MS to obtain the mechanical seal hole defect. Figure 5 shows the MS hole fault (MSH).

### 3.3. Impeller Fault

Seven impellers were used in this study. Out of those seven, six were defect-free, and the seventh was made faulty by removing a portion. The dimensions of the flaw created had a diameter of 2.5 mm, a length of 18 mm, and a depth of 2.8 mm. This provided our study with a faulty impeller condition. Figure 6 shows this fault.

The vibration signals collected from the CP under faulty and normal operating conditions are shown in Figure 7. Figure 8 and Figure 9 show the frequency spectrums of the CP under normal, impeller fault, MSH fault, and MSS fault conditions. The defect frequencies for the impeller fault were calculated using the mathematical equations provided in [2]. It can be seen from Figure 8a,b that the amplitudes of the third, fourth, and fifth harmonics increased because of the impeller defect. Furthermore, some spikes appeared in the frequency spectrum after introducing the impeller defect. The possible reason for this is the interaction of the fluid with the faulty impeller, as can be seen in Figure 8b. In Figure 9, it can be observed that the excitation frequency under mechanical seal defects increased twice with respect to that of the normal condition. The excitation frequency is highlighted in Figure 9a–c.

## 4. Technical Background

### 4.1. Vibration Signals Representation Using Kurtograms

The vibration data obtained from the CP testbed were noisy, and because of different faults, they presented impulses, non-stationarities, and non-linearities. The TD, FD, and TFD techniques lagged in extracting the discriminant parts from the signal because of the above-mentioned reasons. Thus, we used fast kurtograms to clearly depict the impulses, non-stationarities, and non-linearities in the vibration sequences.

The fast kurtogram was first proposed by Antoni [22,23]. A kurtogram is a spectral analysis tool based on spectral kurtosis used to detect non-stationarities and non-linearities in signals. Spectral kurtosis was presented for the first time by Dwyer in 1983 [24,25]. This tool uses power spectral density to detect and show the transients in vibration signals. The main idea on the basis of spectral kurtosis is to have a kurtosis for all frequency values to detect and locate the hidden transient portions of a signal in the frequency domain.

Random processes in vibration signals, such as non-stationary *x(n)* with zero-mean, are split using the given Wold–Cramer equation used in the study [23].
(1)$$X(n)=\int_{-\frac{1}{2}}^{+\frac{1}{2}}H\left(n,f\right)e^{j2\pi \mathrm{fn}}dZ_{x}\left(f\right)$$

$H\left(n,f\right)$ indicates a complex kind of envelope for *x(n)* with frequency f, and $dZ_{x}$ indicates an orthonormal spectral change.

The mathematical definition of spectral kurtosis is given as:(2)$$K_{x}\left(f\right)=\frac{\langle \left|H{\left(n,f\right)}^{4}\right|\rangle }{{\langle \left|H\left(n,F\right)\right|\rangle }^{2}}-2$$

The above-mentioned mathematical definition has important properties; one related to this study is given in Equation (3) [22], as explained below.

In a non-stationary process, when dealing with an impulsive signal *x(n)* having a stationary additive noise *s(n)*, the spectral kurtosis of the calculated measurement $y\left(n\right)=x\left(n\right)+s\left(n\right)$ can be:(3)$$k_{y}\left(f\right)=\frac{k_{x}\left(f\right)}{{\left[1+p\left(f\right)\right]}^{2}}$$

$k_{x}\left(f\right)$ is the spectral kurtosis of $x\left(n\right),$ and $p\left(f\right)$ is the SNR as a function of frequency.

In this study, when the fast kurtogram was applied to real vibration signals, it produced a graph that displayed the spectral kurtosis of different frequency bands in the frequency domain [23]. The kurtograms computed from the CP vibrational sequences clearly showed different color regions when changing the CP operating conditions. Figure 10 depicts the kurtogram images of the vibration sequences for each CP operating conditions. The colors in each block of the kurtogram image represents a portion of the frequency, and the block width represent the bandwidth. The appearance of high-intensity colors in Figure 10b–d indicates the impulses that occurred because of different faults in the CP. These high-frequency impulses were used for fault identification by the DL models utilized in this work. The computation of these colors and bands were based on spectral kurtosis.

### 4.2. Convolutional Encoder with Contrastive Learning

An autoencoder is an NN comprising three main layers: an input layer that takes data as an input, a hidden layer that processes the data, and finally an output layer that provides the output. The first two layers form the encoder, while the last two layers form the decoder. The encoder learns the important representations of the input data, and the decoder performs the reconstruction of the data using the representations learnt by the encoder. The process of data compression and reconstruction by an autoencoder is explained below.

The encoder obtains the input data as **x**, and the hidden layer transforms them into hidden representations **h**.
(4)$$h=a\left(W_{1}x+b_{1}\right),$$

Here, $w_{1}$ and $b_{1}$ show the weight vectors and bias vectors of the hidden layers, while a is the activation function. The decoder reconstructs the input data **x** using the learnt hidden representations **h** with the following equation:(5)$$x_{r}=a\left(w_{2}h+b_{2}\right),$$

$x_{r}$ denotes the reconstructed input data of the autoencoder, and the weight vectors and bias vectors of the output layer are represented by $w_{2}$ and $b_{2}$, respectively. We did not need to reconstruct the data, so we only needed the encoder

The encoder portion of the autoencoder has inherent function of data compression. To achieve the best latent representation of the input data, we modified the data compression process by introducing a series of convolutional layers backed by pooling layers. Therefore, in this paper, we used the convolutional encoder for feature extraction, rather than as a traditional encoder.

The CE comprised convolutional layers, pooling layers, and a fully connected layer. In this study, the CE had the kurtogram images computed from the vibration signals of the CP as its input; from these images, the CE extracted useful features by performing the convolution and pooling processes. The operation of the CE is described below.

The convolutional layers performed the convolution of the kurtograms using various filters, and every channel of the kurtograms underwent the convolution separately. At the end of the convolution operation, several features were obtained through the activation function. The following equation shows the operation of the convolutional layer.
(6)$$x_{c}^{n}=a^{n}\;\left(\sum_{k=1}^{k^{n-1}}w_{k,c}^{n}*x_{k}^{n-1}+b_{c}^{n}\right)$$

Here, n  denotes the order of the current convolutional layer, ∗  shows the convolution of the channel $k=1,\ldots ,k^{n-1}$ for the input $x_{k}^{n-1}$ of the convolutional layer, and $w_{k,c}^{n}$ represents the Cth filter weights in layer $n.\;b_{c}^{n}$ is the bias of the filter c in the convolutional layer n, and $a^{n}\left(.\right)\;$ is the non-linear activation function to extract the features. In this study, we used the (ReLU) as an activation function.

After the convolution process, the pooling process started. The aim of the pooling process is to obtain key features from the feature map obtained from the convolutional layer, hence reducing the amount of processing data. Consequently, the encoder consumes less time and memory to operate on the features data. The output of the pooling layer is given below.
(7)$$x_{c}^{n}=\beta_{c}^{n}down\left(x_{c}^{n-1}\right)+b_{c.}^{n}$$

The $down\left(.\right)$ indicates the down-sampling process, $x_{c}^{n}$ shows the pooling layer’s outcome, and $x_{c}^{n-1}$ the previous layer’s output and the current pooling layer’s input. $\beta_{c}^{n}$ represents a multiplicative bias, while $b_{c}^{n}$ is the additive bias. mMx-pooling was used for size reduction of the features obtained at each convolution operation. The architecture used for the CE is presented in Table 2.

A number of convolution and pooling steps completed the feature extraction from the kurtograms fed to the CE. Cumulatively, four convolutional layers having eight 3 × 3 filters followed by four pooling layers each having eight 2 of 2 × filters, one flattening layer having 512 nodes, and a reshape layer completed the architecture of the CE. An activation function RELU was placed at the out-gate of each convolutional layer. To train the CE, a supervised contrastive loss was used, which kept the features bound to the specific class label based on the contrast in the features. During CE training, the representations learned were propagated to a projection network where the supervised contrastive loss was computed and optimized.

As obvious form the word “contrast”, contrastive learning aims to learn low-dimensional representations of data by contrasting similar and dissimilar samples. Specifically, contrastive learning brings similar samples near each other in the feature pool and pushes the different samples away on the basis of the Euclidean distance [26]. To complete this task, contrastive loss is used, which explicitly shows how close the features are based on their similarity. In contrast to other common losses, such as cross-entropy, mean-squared error, etc., whose goal is to predict the labels directly, contrastive loss targets train the semantically useful feature representations of the data. Contrastive learning has shown promising results in dealing with computer vision in both supervised [27,28] and unsupervised ways [29,30]. Franceschi et al. [31] also applied contrastive learning successfully to time series data. In this study, we used contrastive learning for fault diagnosis in CPs. Generally, in contrastive learning, the projection network $Proj\left(.\right)$ obtains the features from an encoder and maps those features vector x to a vector $z=Proj\left(x\right)\in R^{D_{CE}}$. The projection network is instantiated as a multi-layer perceptron having only a hidden layer with size of 2048 and an output vector with size of 128. The output of this network is normalized to a unit hypersphere, which allows the use of the inner product for measuring distances in projection space. This projection network is discarded at the end of contrastive training, and eventually, the inference model obtains exactly the same parameters as the output of a convolutional encoder. This is how contrastive learning works. There are several settings in which contrastive learning can be applied, e.g., supervised and unsupervised contrastive learning. However, in this study, we considered multiclass labelled data. Therefore, for differentiating data and grouping data by features to their respective class label, we needed a setup of supervised contrastive learning in the projection network

For supervised learning, the general mathematics of contrastive loss is incapable of measuring the distance in the projection space because of multiclass labels. As a result, the following equations are used for supervised inference [21].
(8)$$l_{out}^{sup}=\sum_{i\in I}l_{out,i}^{sup}=\sum_{i\in I}\frac{-1}{\left|P\left(i\right)\right|}\sum_{p\in P\left(i\right)}\log\frac{\exp(s_{i}\cdot s_{p}/\tau )}{\sum_{a\in A\left(i\right)}\exp(s_{i}\cdot s_{a}/\tau )}$$
(9)$$l_{in}^{sup}=\sum_{i\in I}l_{in,i}^{sup}=\sum_{i\in I}-\log\{\frac{1}{\left|P\left(i\right)\right|}\sum_{p\in P\left(i\right)}\frac{\exp(s_{i}\cdot s_{p}/\tau )}{\sum_{a\in A\left(i\right)}\exp(s_{i}\cdot s_{a}/\tau )}\}$$

$P\left(i\right)=\left\{p\in A\left(i\right)\;:y_{p}=y_{i}\right\}$ points to the indices set consisting of all positives in the features data extract by the CE. In Equation (8), the summation is outside the log $\left(l_{out}^{sup}\right),$ while in Equation (9), the summation is inside the log $\left(l_{in}^{sup}\right)$. Both losses have properties such as generalization to an arbitrary number of positives, a contrastive power increase with more negatives, and the intrinsic ability to perform hard positive/negative mining [21].

### 4.3. Fault Identification Using a Linear Classifier

After completion of the supervised contrastive inference in the projection network, the weights of the CE were kept frozen, and a linear classifier was trained which had a hidden layer and an output layer. The weights of the fully connected layer that had a SoftMax function were optimized. Because this was a multiclass classification problem, a categorical cross entropy loss was used to perform the classification. The vibration sequences were transformed to kurtograms, from which significant features were extracted by the CE. These features were segregated through supervised contrastive learning and were classified by the linear classifier. Figure 11 shows the data flow with architecture details of the models used for the supervised contrastive learning-based CP fault diagnosis.

## 5. Results and Discussion

For the evaluation of the performance of our method for fault diagnosis, it was necessary to organize a proper training and testing subsets of the data. This study considered a total of 1200 kurtograms, 300 for each class. The kurtograms were split randomly at a ratio of 8:2. The training set comprised 960 kurtograms, and the testing set contained 240 data samples.

### Performance and Comparison

In this study, we computed the fast kurtograms of vibration signals and then used a convolutional encoder to extract features from the kurtograms. The extracted features were provided to a projection network where the contrastive loss was optimized to infer the contrast in the data features. Once the interclass contrast in the data was inferred, the weights of the CE were kept frozen, and a classifier was trained using the learned representations. The classifier then identified the health condition of the CP. The proposed model showed the best results in terms of accuracy and robustness. To compare the performance of our method with those of the reference methods, we used different metrics, i.e., accuracy, precision, recall, and F1 score. The equations for the calculation of these metrics are presented below.
(10)$$Accuracy=\frac{\sum_{\alpha }^{E}TP_{\alpha }}{N}\times 100\%\;$$
(11)$$Precision=\frac{\sum_{\alpha }^{E}n_{\alpha }\times \left(\frac{TP_{\alpha }}{TP_{\alpha }+FP_{\alpha }}\right)}{N}\times 100\%$$
(12)$$Recall=\frac{\sum_{\alpha }^{E}n_{\alpha }\times \left(\frac{TP_{\alpha }}{TP_{\alpha }+FN_{\alpha }}\right)}{N}\;\times \;100\%\;$$
(13)$$F1=\frac{1}{N}\sum_{\alpha }^{E}n_{a}\times 2\times \sum_{\alpha }^{E}\left(\frac{Recall_{\alpha }\times Precision_{\alpha }}{Recall_{\alpha }+Precision_{\alpha }}\right)\times 100\%$$

Here, TPs are true positives, FPs are false positives, and FNs denote false negative outcomes of the features of class a. $n_{a}$ is the total number of samples of class a. E depicts the number of classes, and N is number of data samples in the test data.

Shao Haidong et al. [32] used a deep autoencoder features learning method for rotatory machines fault diagnosis. The data used for fault diagnosis were acquired using a vibrations sensor; to avoid signal processing, these authors directly developed a deep autoencoder and then followed the framework provided in a previous methodology [32]. They obtained good results in fault diagnosis, which led us to compare their method to our proposed method. For a fair comparison between the referenced and the proposed fault diagnosis frameworks, we applied the deep autoencoder features learning method to our CP dataset. The referenced method extracted features without preprocessing using deep auto encoders, because it could not display the transients in the vibration signals. In contrast, the proposed method used fast kurtograms to explicitly show the impulses, non-stationarities, and non-linearities in vibration data because of faults in the CP. Therefore, it was easy for the pipeline to identify the faults with high accuracy. Furthermore, in the referenced method, the features learned by the deep autoencoder were directly fed to a SoftMax classifier for fault classification, while the proposed method used a convolutional encoder to extract features and fed them to a projection network for the optimization of a contrastive loss, to learn the contrasting features in the data. To keep these contrasting features as a base for the purpose of classification, the weights of the CE and the contrastive features were kept frozen, and a classifier was trained to infer the contrast and decide about the state of the CP’s health. The final results showed that the proposed model outperformed the referenced method, i.e., the proposed method had 99.1% accuracy, while the referenced method had 90.4% accuracy.

To demonstrate the superiority of our method, we compared our method with another framework that used a sparse autoencoder based on multivariate data to detect faults. This study includes an offline detection and an online detection. Because our study was based on offline fault detection, we applied the offline setup to our data and obtained good results in fault detection; however, the method did not surpass the proposed technique for CP health state identification. The proposed method outperformed the referenced method in identifying the conditions of the CP. To statistically compare the performances of the proposed and referenced methods, we used the confusion matrices shown in Figure 12, which clearly proved the proposed method’s superiority with respect to the referenced methods.

The performance of our method compared to those of the other described methods is shown in Table 3. The stats clearly shows that our method performed better than the referenced methods in fault identification. The proposed method classification results had an accuracy of 99.1%, a precision of 98.95%, a recall of 99%, and an F1 score of 98.75% using the 3.0 bar inlet pressure vibration data. The results are explained below.

The proposed model also showed stability when the experiment was repeated 10 times; the model had higher accuracy than the referenced methods because we used contrastive learning that provided the classifier with highly discriminant features, while the other methods provided non-discriminant and indifferentiable features to the classifier, which resulted in low accuracy of the CP dataset. Figure 13 presents the plots of features obtained from the principal component analysis of the proposed and referenced models before the classification.

To prove the robustness of our model, we trained the model on a dataset with a 3.0 bar inlet pressure and then tested it on the CP vibration data with a 3.5 bar pressure at the inlet. The proposed model provided promising fault classification results even on a different dataset; the accuracy was 98.8%, the precision was 98.1%, the recall was 97.9%, and the F1 score was 98%.

## 6. Conclusions

This paper proposes a novel and robust framework for CP fault diagnosis based on kurtograms and contrastive learning. The proposed method involved three major steps. First, vibration signals of four different CP health conditions were considered to pre-process the signals and explicitly show discriminant fault features with the fast kurtogram. Second, these kurtograms were fed to a convolutional encoder to extract features using the convolution and pooling operations; furthermore, these features were provided to a projection network, where the contrastive loss was optimized to infer the contrast in the extracted features. In the third phase, the inference of the projection network was used to determine the health of the CP; the weight of the encoder and the learning in the projection network were maintained, and a classifier was trained. The proposed model yielded very promising results in fault classification, with an accuracy of 99.1%; however, we need to test the model on hydraulic CP faults. We will extend our research to classify hydraulic CP faults, such as cavitation, using the proposed method. Besides this, we will widen the scope of the proposed approach by applying it to pumps data that have missing labels.

## Figures and Tables

**Figure 1 sensors-22-06448-f001:**
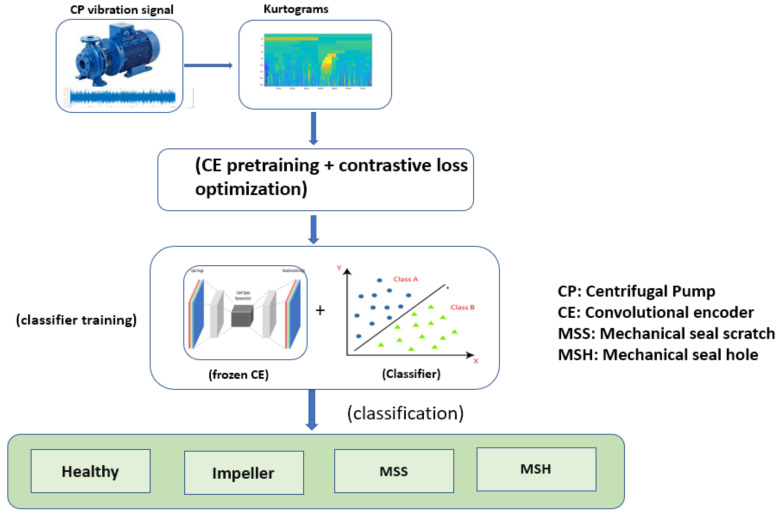
Flow chart of the proposed approach used for CP fault detection.

**Figure 2 sensors-22-06448-f002:**
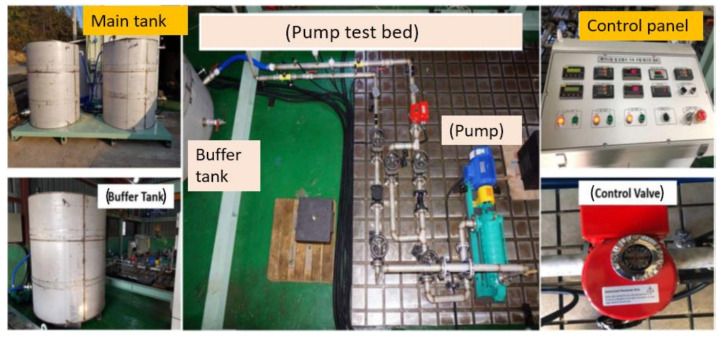
Experimental setup utilized for vibration data acquisition.

**Figure 3 sensors-22-06448-f003:**
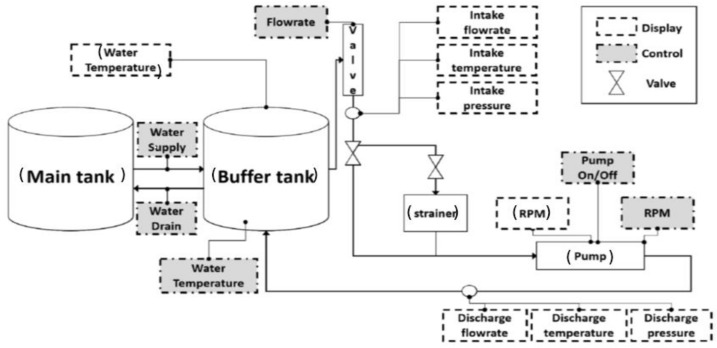
Block scheme of the experimental testbed.

**Figure 4 sensors-22-06448-f004:**
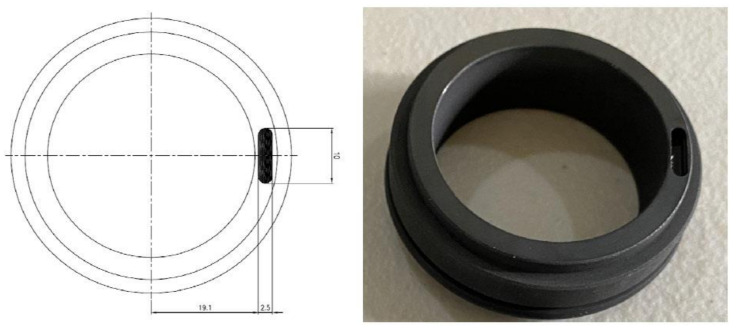
Mechanical seal scratch fault.

**Figure 5 sensors-22-06448-f005:**
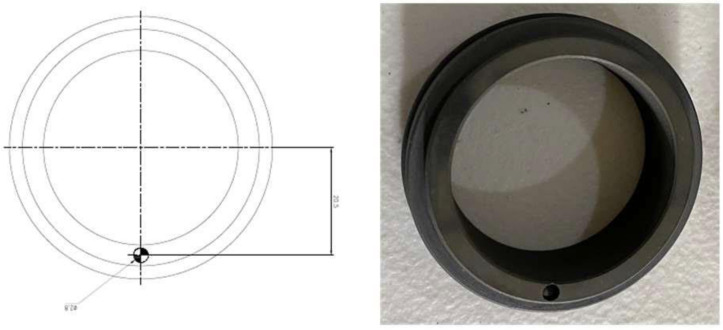
MS hole fault.

**Figure 6 sensors-22-06448-f006:**
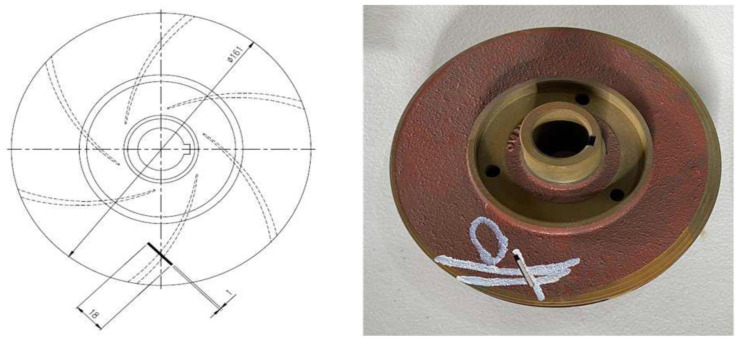
Impeller defect.

**Figure 7 sensors-22-06448-f007:**
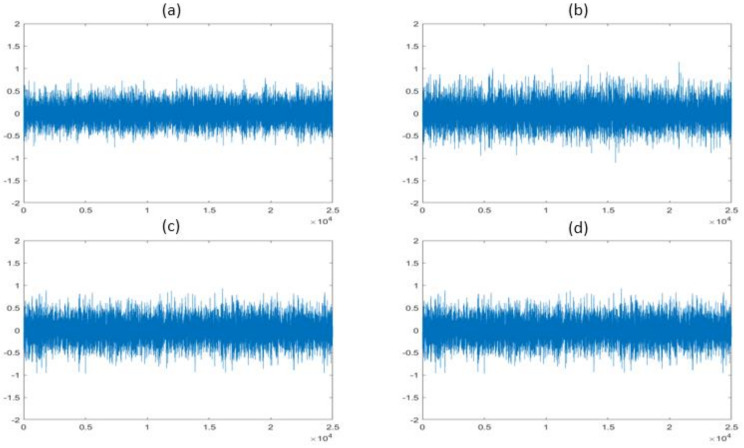
Time-domain vibration sequences in (**a**) normal, (**b**) impeller fault, (**c**) MSH fault, and (**d**) MSS fault conditions.

**Figure 8 sensors-22-06448-f008:**
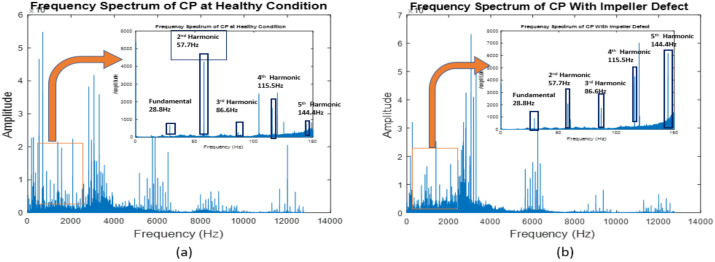
Frequency spectra of the CP under (**a**) normal and (**b**) impeller defect conditions.

**Figure 9 sensors-22-06448-f009:**
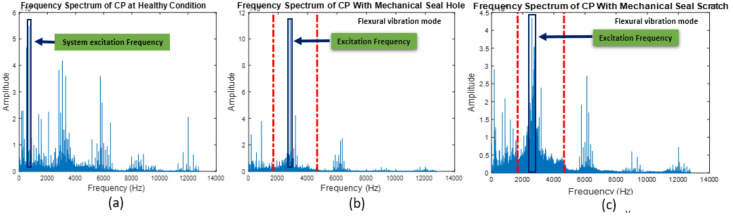
Frequency spectrum of the CP. (**a**) Healthy conditions, (**b**) MSH fault, and (**c**) MSS fault.

**Figure 10 sensors-22-06448-f010:**
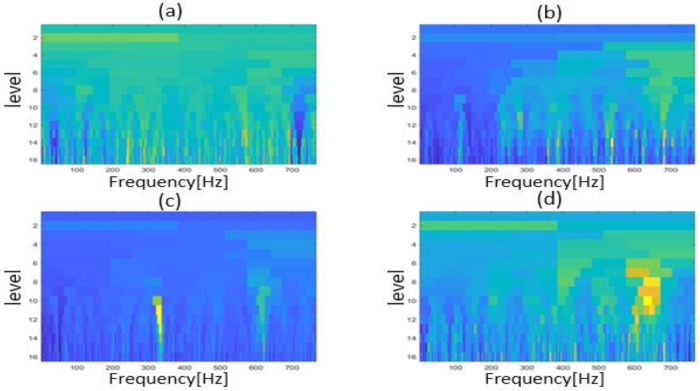
Kurtogram patterns computed for the signals representing (**a**) normal conditions, (**b**) MSS fault, (**c**) MSH fault, and (**d**) impeller defect under a 3.0 bar pressure.

**Figure 11 sensors-22-06448-f011:**
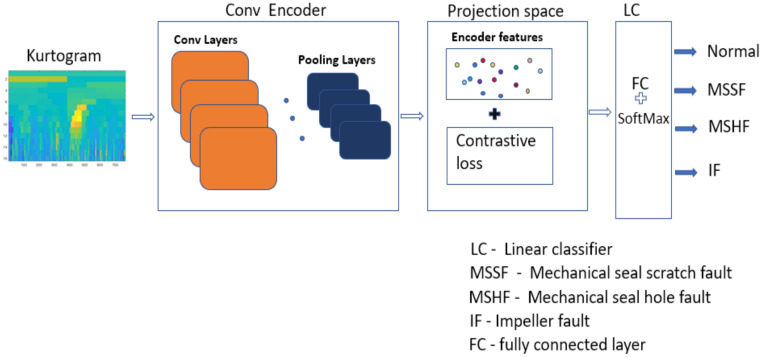
Flow diagram of feature extraction, contrastive learning, and classification.

**Figure 12 sensors-22-06448-f012:**
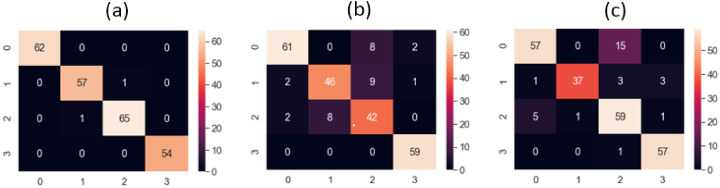
Confusion matrices (CM) of (**a**) the proposed method, (**b**) the DAE feature learning method, and (**c**) the SAE-based scheme for a 3.0 bar pressure vibration dataset.

**Figure 13 sensors-22-06448-f013:**
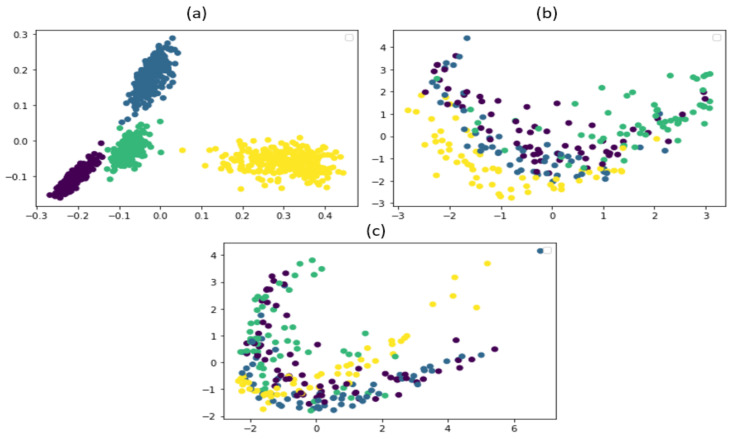
PCA feature plots of the (**a**) proposed method, (**b**) the SAE-based fault detection scheme, and (**c**) the DAE feature learning method.

**Table 1 sensors-22-06448-t001:** Attributes of the testbed for data acquisition.

Tool	Specification
Accelerometer (622b01)	0.42–10 kHz frequency range 100 mV/g (10, 2 mV/(m/s^2^)) ± 5% of sensitivity
DAQ system (NI 9234)	0–13.1 MHz range of frequency generator having four input analogue channels with a 24-bit resolution power

**Table 2 sensors-22-06448-t002:** Architecture of the convolutional encoder.

Layer Number	Filters Number	Kernel	Output	Activation ftn
1, Conv.	8 Filters	3 × 3	128 × 128 × 8	ReLU
2, Maxpool	8 Filters	2 × 2	64 × 64 × 8	-
3, Conv.	8 Filters	3 × 3	64 × 64 × 8	ReLU
4, Maxpool	8 Filters	2 × 2	32 × 32 × 8	-
5, Conv.	8 Filters	3 × 3	32 × 32 × 8	ReLU
6, Maxpool	8 Filters	2 × 2	16 × 16 × 8	-
7, Conv.	8 Filters	3 × 3	16 × 16 × 8	ReLU
8, Maxpool	8 Filters	2 × 2	8 × 8 × 8	-
9, Flatten	512 Nodes	-	512	
10, Reshape	-	-		

**Table 3 sensors-22-06448-t003:** Results obtained from the proposed and compared methods for 3.0 bar inlet pressure vibration data.

Metric	Proposed	SAE Based Fault Detection Scheme	DAE Feature Learning Method
Accuracy	99.1%	92.75%	90.4%
Recall	98.95%	91.4%	89.5%
Precision	99%	90.2%	91%
F1 score	98.75%	90.8%	90.5%

## Data Availability

Data will be provided upon request.

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
