# Peer review of "A Centrifugal Pump Fault Diagnosis Framework Based on Supervised Contrastive Learning"

_sensors, 2022, doi:10.3390/s22176448_

Round 1
Reviewer 1 Report
Q1: In introduction, author should provide the summary about the fault diagnosis of pump and AI, which has some significant research papers that can not used as basis and comparison for this paper. ref1:Multi-fault Condition Monitoring of Slurry pump with Principle Component Analysis and Sequential Hypothesis Test.
Q2:On line 78-79, in this study, "Kurtogram is used to visualize the transients in the vibration signal " is not exactly corrct. Kurtosis is probability index that can not visualize the transient signal in the vibration signals.
Q3:Figures 1 and 7 are not clear.
Q4: Table 2 need more explanations.
Q5: In 4.3, the procedure about the Figure 9 should be added.
Q6: the picture quality for Figure 11 is poor.
Reviewer 2 Report
Title: A Centrifugal Pump Fault Diagnosis Framework based on Supervised Contrastive Learning
Journal: Sensors
Decision: accepted with minor revision
General comments:
This reported manuscript proposed a method that can make fault diagnosis for the centrifugal pump, this method is composed of kurtogram image, deep learning tool-convolutional encoder. It can deal with the non-stationary signal effectively and be validated with a real industrial testbed. It is suggested to be accepted with minor revision. However, some detailed problems need to be clarified:
1. In the title, the authors used the word ‘framework’, how do define the framework? It is suggested the authors revise the title and choose the word cautiously.
2. Have the authors verified this proposed method could also be reliable to the other centrifugal pump?
3. The authors used ‘high classification accuracy in the abstract, what is the exact value? What is the error?
4. From line 107 to line 116, the term ‘convolutional encoder’ and ‘CE’ are both used, it is better to keep them consistent.
5. The quality of some figures should be improved and the resolution is low.
6. It is suggested to plot the FFT figure in Fig7 to tell the difference for various fault types.
Reviewer 3 Report
This paper deals with an algorithm for fault diagnosis of Centrifugal pump. The method uses vibration signals data obtained from a CP. First, kurtogram images of time series vibration sequences are computed, then a convolutional encoder (CE) with a supervised contrastive loss is used to extract the discriminant features related to faults from the kurtogram images. The convolutional encoder is pre-trained on the kurtograms with the supervised contrastive loss to infer the contrasting features belonging to different CP data classes. The learned representations of the convolutional encoder are kept as it is, and a linear classifier is trained above the frozen convolutional encoder, which completes the fault identification.
The application support used to validate the method is relevant as centrifugal pumps are key elements in several industrial fields, and these are elements whose failure rates are high, so it is useful to propose robust algorithms to monitor them.
Convolutional encoders are powerful deep learning tools that are successfully used in several fields, including fault diagnosis. They have the advantage of performing with a single tool all the synthesis stages of a classifier (extraction, selection and classification). It is therefore a wise choice. The Kurtograms computation and the use of a supervised contrastive loss function can be considered as a new contribution.
Experimental results, obtained on recorded data show the effectiveness of the proposed approach.
however, the paper can be improved on the following points:
1) In the abstract, the authors mentioned that “The vibration signals data obtained from a CP is non-stationary because of the impulses caused by different faults, thus a traditional domain analysis, e.g., time domain and frequency domain, is not the best option to pre-process the non-stationary signals “. If the processing method proposed by the authors is neither in the time domain nor in the frequency domain, it is in which domain?
2) Supervised learning assumes the availability of a labeled matrix which associates each set of data with a class of normal functioning or faulty functioning. But in reality, the databases are rarely complete. What is the expected performance of the algorithm in these cases?
3) The proposed method achieves an accuracy of 99.1%, which is a very good performance. There are decision-making methods in the literature that make it possible to achieve 100% online accuracy by replacing point-in-time decision-making by decision-making over an observation window, it would be good to quote these methods in the introduction, for example, A temporal-based SVM approach for the detection and identification of pollutant gases in a gas mixture. Applied Intelligence.
4) The introduction can be enhanced by review papers on artificial intelligence tools applied to failure diagnosis such as: Machine Learning and Deep Learning Algorithms for Bearing Fault Diagnostics -A Comprehensive Review. As well as hybrid methods that combine physical models (such as bond graphs) to generate missing data and artificial intelligence for diagnosis.
Round 2
Reviewer 1 Report
In the introduction, the paper should compared the existing papers with the authors' paper, which shows the innovation and new idea in the paper. It is suggested that author should have provide one paragraph to summarize the researches about the fault diagnosis of pump to give the advantages in method, idea etc. that is used in the paper.
